# High Levels of TRIM5α Are Associated with Xenophagy in HIV-1-Infected Long-Term Nonprogressors

**DOI:** 10.3390/cells10051207

**Published:** 2021-05-14

**Authors:** Fabiola Ciccosanti, Marco Corazzari, Rita Casetti, Alessandra Amendola, Diletta Collalto, Giulia Refolo, Alessandra Vergori, Chiara Taibi, Gianpiero D’Offizi, Andrea Antinori, Chiara Agrati, Gian Maria Fimia, Giuseppe Ippolito, Mauro Piacentini, Roberta Nardacci

**Affiliations:** 1Department of Epidemiology, Preclinical Research and Advanced Diagnostics, National Institute for Infectious Diseases, Lazzaro Spallanzani-IRCCS, 00149 Rome, Italy; fabiola.ciccosanti@inmi.it (F.C.); marco.corazzari@uniupo.it (M.C.); rita.casetti@inmi.it (R.C.); alessandra.amendola@inmi.it (A.A.); dilettacollalto@alice.it (D.C.); giulia.refolo@inmi.it (G.R.); chiara.agrati@inmi.it (C.A.); gianmaria.fimia@inmi.it (G.M.F.); giuseppe.ippolito@inmi.it (G.I.); mauro.piacentini@uniroma2.it (M.P.); 2Department of Health Sciences, University of Piemonte Orientale, 28100 Novara, Italy; 3Center for Translational Research on Autoimmune and Allergic Disease, School of Medicine, University of Piemonte Orientale, 28100 Novara, Italy; 4Clinical Department, National Institute for Infectious Diseases, Lazzaro Spallanzani-IRCCS, 00149 Rome, Italy; alessandra.vergori@inmi.it (A.V.); chiara.taibi@inmi.it (C.T.); gianpiero.doffizi@inmi.it (G.D.); andrea.antinori@inmi.it (A.A.); 5Department of Molecular Medicine, Sapienza University of Rome, 00185 Rome, Italy; 6Department of Biology, University of Rome “Tor Vergata”, 00133 Rome, Italy

**Keywords:** autophagy, TRIM5α, HIV-1, long-term nonprogressors, xenophagy

## Abstract

Autophagy is a lysosomal-dependent degradative mechanism essential in maintaining cellular homeostasis, but it is also considered an ancient form of innate eukaryotic fighting against invading microorganisms. Mounting evidence has shown that HIV-1 is a critical target of autophagy that plays a role in HIV-1 replication and disease progression. In a special subset of HIV-1-infected patients that spontaneously and durably maintain extremely low viral replication, namely, long-term nonprogressors (LTNP), the resistance to HIV-1-induced pathogenesis is accompanied, in vivo, by a significant increase in the autophagic activity in peripheral blood mononuclear cells. Recently, a new player in the battle of autophagy against HIV-1 has been identified, namely, tripartite motif protein 5α (TRIM5α). In vitro data demonstrated that TRIM5α directly recognizes HIV-1 and targets it for autophagic destruction, thus protecting cells against HIV-1 infection. In this paper, we analyzed the involvement of this factor in the control of HIV-1 infection through autophagy, in vivo, in LTNP. The results obtained showed significantly higher levels of TRIM5α expression in cells from LTNP with respect to HIV-1-infected normal progressor patients. Interestingly, the colocalization of TRIM5α and HIV-1 proteins in autophagic vacuoles in LTNP cells suggested the participation of TRIM5α in the autophagy containment of HIV-1 in LTNP. Altogether, our results point to a protective role of TRIM5α in the successful control of the chronic viral infection in HIV-1-controllers through the autophagy mechanism. In our opinion, these findings could be relevant in fighting against HIV-1 disease, because autophagy inducers might be employed in combination with antiretroviral drugs.

## 1. Introduction

Autophagy is the major intracellular degradation mechanism in addition to the ubiquitin-proteasome system, and it is a key physiological process for eukaryotic cell homeostasis [1]. Among the most important functions of autophagy, in addition to maintaining cell homeostasis, its role is in host defense through several biological functions: the direct elimination of invading pathogens, control of adaptive immunity through the regulation of antigen handling and presentation, induction of innate immune memory or trained immunity, and the modulation of inflammation [2,3,4,5,6]. Pathogens have evolved a series of strategies to inhibit the immunity-supporting roles of autophagy and to hijack autophagy protein activities for their own benefit [7]. Concerning the role of autophagy in HIV-1 infection, it has been demonstrated that it is involved in both HIV-1 replication and pathogenesis [8,9].

Literature data show that several members of the TRIM (tripartite motif protein) protein family are regulators of both viral- and non-viral-induced autophagy [10,11]. Among the TRIM family members, TRIM5α has emerged as a key factor in the autophagy mechanism. Numerous studies have documented the association of TRIM5α with some autophagy-related proteins, namely, ULK1, BECN1, ATG5, ATG16L1, LC3s/GABARAPs, ATG14, AMBRA1 and SQSTM1 [12,13,14,15,16,17]. TRIM5α was shown to potently block the ability of HIV-1 to infect cells from certain species of Old World monkeys, acting as an upstream regulator of autophagy by providing a platform for the assembly of activated ULK1 and BECN1 as well as a receptor for selective autophagy. In its role as an autophagic cargo receptor, TRIM5α directly recognizes viral capsid sequences via its SPRY or CypA domains, inducing premature capsid disassembly and virus restriction [18,19,20].

Studies on human TRIM5α demonstrated that this variant was much less efficient in blocking HIV-1 infection than rhesus monkey TRIM5α, thus precluding human TRIM5α for the HIV-1 capsid effective precision autophagy [19,21]. Human TRIM5α restriction efficacy may depend on autophagy processes, as shown in Langerhans cells that are known to be the first line of defense against HIV-1 [14]. Recently published data indicated that autophagy machinery is required for TRIM5α to transduce antiviral signaling. Autophagy provides TRIM5α with a functional platform to initiate an antiviral state that is protective against HIV-1 [22]. Thus, the characterization of the mechanism/s through which TRIM5α restricts HIV-1 is still unknown and needs to be investigated. The aim of the present study was to analyze the expression of TRIM5α in vivo in HIV-1-infected patients. We have focused our attention on LTNP, in which the resistance to HIV-1-induced pathogenesis is accompanied by a significant increase in the autophagic activity in blood cells [23]. Our results suggest a role for TRIM5α in the control of HIV-1 infection through autophagy in LTNP, the special subset of HIV-1-infected patients that spontaneously and durably maintain extremely low virus replication.

## 2. Materials and Methods

### 2.1. Patients and Healthy Donors

Venous blood was collected from adult human subjects that were cared for in the Outpatients HIV clinic of the National Institute for Infectious Disease Lazzaro Spallanzani-IRCCS Hospital, Rome, Italy, and provided written informed consent to participate in the study (Ethics Committee approval no. 49/2010).

Patients enrolled in the study were 10 LTNP, 15 HIV-1-infected normal progressor patients (NP), and 12 healthy donors (HD) of both sexes. Demographic and laboratory findings of our cohort of patients are summarized in Appendix A. LTNP plasma HIV-1 RNA (viral load) median value was 2.4 ± 0.7 log_10_ cp/mL, with CD4 T cells > 500/µL. LTNP were infected for an average of 8 years. NP were not on antiretroviral therapy and the median value of viral load was 3.9 ± 1.4 log_10_ cp/mL and <500 CD4 T cells/µL. HD were HIV-1 negative blood donors. Plasma HIV-1 RNA levels were measured with an Abbott Real-time HIV-1 assay according to the manufacturer’s instructions (Abbott Molecular, RealTime HIV-1). Peripheral blood mononuclear cells (PBMC) were obtained from residual blood samples intended for diagnostic tests. PBMC were isolated by Ficoll/Hypaque (GE Healthcare, 17-1440-02) centrifugation of heparinized blood and utilized immediately for immunocytochemical analysis, and frozen at −80 °C until Western blot, qPCR, flow cytometry and cell sorting were performed.

### 2.2. Immunocytochemistry of TRIM5α

PBMC from 15 NP, 5 LTNP and 8 HD were fixed in 40 mg/mL freshly depolymerized paraformaldehyde (Sigma-Aldrich, Saint Louis, MO, USA, P-6148) in phosphate-buffered saline (PBS, Sigma-Aldrich, Saint Louis, MO, USA, P-4417) pH 7.2. Immunocytochemistry was performed as previously reported [24] on fixed PBMC. The primary antibody utilized was the rabbit polyclonal anti-TRIM5α (1:50, Novusbio NBP1-76601).

The percentage of positive PBMC/total PBMC, was counted for TRIM5α staining. Three independent observers evaluated the number of positive cells by using a light microscope without the knowledge of clinical diagnosis. A minimum of 500 PBMC/patient were analyzed.

### 2.3. Fluorescence-Activated Cell Sorting (FACS) Analysis

We performed FACS analysis on PBMC from 4 NP, 4 LTNP and 7 HD. The following anti-human monoclonal antibodies were used for staining surface and intracellular antigens: Fluorescein isothiocyanate (FITC)-conjugated anti-CD45, phycoerythrin (PE)-conjugated anti-CD14, allophycocyanin (APC)-conjugated anti-CD19, Horizon V450-conjugated anti-CD4, APC-Cyanine7 (Cy7)-conjugated anti-CD8, APC-H7-conjugated anti-CD14, Horizon V500-conjugated anti-CD3, Horizon V500-conjugated anti-CD45, Peridinin Chlorophyll Protein Complex (PerCP)-conjugated anti-CD3, PerCP-conjugated anti-7AAD, all purchased from BD Pharmingen (San Diego, CA, USA), Alexa Fluor 488-conjugated anti-TRIM5 (D-6) purchased from Santa Cruz Biotechnology (Santa Cruz Biotechnology, CA, USA), and APC-VIO770-conjugated anti-CD8 purchased from Miltenyi Biotec S.r.l (Bologna, Italy). The isotype-matched antibodies were purchased from the corresponding companies. Frozen PBMC were stained with a cocktail of surface antibodies prepared in PBS, 10 mg/mL BSA, 1 mg/mL NaN_3_ buffer, containing PE-CD14, APC-CD19, APC-VIO770-CD8, Horizon V450-CD4 and Horizon V500-CD3. PerCP-7AAD was also added to test cell viability. After 15 min of incubation at 4 °C, cells were washed twice and fixed with 1% formalin. For intracellular staining, Alexa 488-TRIM5 monoclonal antibody prepared in PBS, 10 mg/mL BSA, 1 mg/mL NaN_3_, 0.5% saponin buffer was added. After 20 min at room temperatures, cells were twice washed. Labeled-cells were acquired with an FACS Canto II flow cytometer (BD Biosciences, San Jose, CA, USA) and data analysis was performed using FlowJo software version 9.3.2 (Tree Star, Ashland, OR, USA) and Kaluza analysis software (Beckman Coulter, Milano, Italy).

### 2.4. Immunofluorescence

For immunofluorescence experiments, PBMC from 5 NP, 5 LTNP and 2 HD were incubated with the following antibodies: mouse monoclonal anti-LC3B (1:50, Cosmo Bio, CTB-LC3-1-50); rabbit polyclonal anti-TRIM5α (1:50, Novusbio NBP1-76601), human monoclonal anti-gp120 (kindly provided by Dr. Jean-Luc Perfettini, Paris, France); mouse monoclonal anti-HIV-1 Nef (1:50, Abcam ab42355).

Sections were thoroughly rinsed with PBS, then incubated for 1 h at RT with 1:400 Alexa 594 conjugated goat anti-rabbit IgG (Invitrogen, Rockford, IL, USA, A-11037); 1:400 Alexa 488 conjugated goat anti-mouse IgG (H + L) (Invitrogen, Life Technologies, A-11029), Alexa 647 1:400 goat anti-rabbit IgG (Invitrogen, Life Technologies, A32733), 1:400 Alexa 488 conjugated goat anti-human IgG (H + L) (Invitrogen, Life Technologies, A-11013). Controls were performed by omitting the primary antibodies. Slides were observed and photographed in a Leica TCS SP2 confocal microscope (Leica Microsystems GmgH, Ernst-Leitz-trasse 17-37 35578 Wetzlar, Germany).

The percentage of colocalizations between gp120-positive dots, with LC3-positive dots and TRIM5α-positive dots with respect to the total gp120-positive dots was evaluated. 

A minimum of 30 cells/patient were observed. Cell counting was performed by 3 independent researchers; data are presented as mean ± S.D.

### 2.5. Western Blot Analysis

Total proteins were extracted from PBMC isolated from 7 NP, 6 LTNP and 7 HD, isolated by Ficoll-Hypaque, by using the Nuclear Extract Kit (Active Motif, Carlsbad, CA, USA, 40010) and resolved by electrophoresis through 4–12% Bis-Tris Plus gel (Invitrogen, Rockford, IL, USA, NW04120BOX) and electroblotted onto nitrocellulose (Bio-RadLaboratories, Hercules, CA, USA, 170-4158) membranes. Blots were incubated with 50 mg/mL non-fat dry milk in buffered saline with Tween-20 (TBST) and then in primary antibodies diluted in 50 mg/mL non-fat dry milk in TBST, overnight, at 4 °C. Primary antibodies were: rabbit polyclonal anti-TRIM5α (1:1000, Novusbio, Centennial, CO, USA, NBP1-76601), anti-glyceraldehyde-3-phosphate dehydrogenase (GAPDH, 1:60,000, Calbiochem, CM1001). Detection was achieved using the specific horseradish peroxidase-conjugate secondary antibody (anti mouse IgG 1:5000, Santa Cruz Biotechnology, CA, USA sc-2005; or anti rabbit IgG 1:5000, Santa Cruz Biotechnology, sc-2004) and visualized with Clarity Western ECL (Biorad, 170-5060). Mouse anti-GAPDH antibody was used to monitor equal protein loading. Western blot images were analyzed densitometrically using ChemiDoc Touch Imaging System (Bio-RadLaboratories, Hercules, CA, USA) and processed with ImageJ software in order to quantify the band intensity.

### 2.6. RT-PCR and qPCR

RNA was extracted from PBMC from 4 NP, 8 LTNP and 4 HD by using Trizol reagent (Invitrogen, Rockford, IL, USA, 15596-026) as indicated by the supplier. cDNA was generated using a reverse transcription kit (Promega Italia, Milano, Italy, M5101) according to the manufacturer’s recommendations (RT-PCR). Quantitative PCR (q-PCR) reactions were performed with a Rotorgene 6000 (Qiagen, Hilden, Germany) thermocycler, as previously described [25]. Primer sets for all amplicons were designed using the Primer-Express 1.0 software system (Roche Diagnostics, Indianapolis, IN, USA).
*RPL34* forward: 5′-GTCCCGAACCCCTGGTAATAGA-3′*RPL34* reverse: 5′-GGCCCTGCTGACATGTTTCTT-3′*TRIM5α/**γ* forward: 5′-TACGGCTACTGGGTTATAGG-3′*TRIM5α/**γ* reverse: 5′-CCAACACGATCAGGACAAAA-3′

The cumulative expression levels of both isoforms, alpha and gamma of TRIM5 (TRIM5α/γ), were evaluated at mRNA level because we were not able to discriminate the two isoforms by using an SYBR Green-based qPCR assay.

The mRNA level of the ribosomal protein L34 (*RPL34*) was used as an internal control, and the comparative Ct method (ΔΔCt) was used for relative quantification of gene expression. qPCR efficiency has been estimated by using the free online Primer- and amplicon-specific PCR estimation tool ‘PCR efficiency Calculator’ (http://130.60.24.89/efficiency.html, 10 April 2020). qPCR efficiency for both *RPL34* and *TRIM* was >2.

### 2.7. Electron Microscopy

PBMC from 5 NP and 5 LTNP were fixed with 2.5% glutaraldehyde (Sigma-Aldrich, R1012) in 0.1 M cacodylate buffer, pH 7.4, for 45 min at 4 °C, rinsed in buffer, postfixed in 1% OsO_4_ in 0.1 M cacodylate buffer, pH 7.4, dehydrated, and embedded in Epon resin (Agar Scientific, Essex, UK, 45359-1EA-F). Grids were thoroughly rinsed in distilled water, stained with aqueous 2% uranyl acetate for 20 min, and photographed under a transmission electron microscope, JEOL JEM 2100 Plus (Japan Electron Optics Laboratory Co. Ltd. Tokyo, Japan). Images were captured digitally with a digital camera TVIPS (Tietz Video and Image Processing Systems GmbH. Gauting, Germany).

For immuno-electron microscopy experiments, PBMC from 6 NP and 6 LTNP were fixed in 20 mg/mL freshly depolymerized paraformaldehyde and 0.2% glutaraldehyde in 0.1 M cacodylate buffer, pH 7.4, for 1 h at 4 °C. Samples were rinsed in the same buffer, partially dehydrated, and embedded in London Resin White (Agar Scientific Ltd., Essex, UK, AGR1282). Ultrathin sections were pre-incubated with 10% normal goat serum in 10 mM PBS containing 10 mg/mL BSA and 1.3 mg/mL NaN_3_ (medium A) for 15 min at RT. Sections were incubated with a rabbit polyclonal anti-TRIM5α antibody (Novusbio, Centennial, CO, USA, NBP-76601) diluted 1:25 in medium A overnight at 4 °C. After rinsing in medium A containing 0.01% Tween-20 (Merck, 822,184), sections were incubated with anti-rabbit conjugated to 15 nm colloidal gold (BBInternational, Cardiff, United Kingdom, EM.GAR 15) and diluted 1:30 in medium A containing fish gelatin for 1 h at RT. After incubation with 10% normal goat serum in medium A for 15 min at RT, a second immune-labeling was performed, using a mouse monoclonal anti-HIV-1 Nef (Abcam, Cambridge, MA, USA, ab42355) as a primary antibody and goat anti-mouse IgG conjugated to 5 nm colloidal gold as a secondary antibody (BBInternational, Cardiff, United Kingdom, EM.GAM 5). Grids were thoroughly rinsed in distilled water, contrasted with aqueous 20 mg/mL uranyl acetate for 20 min, and photographed under the transmission electron microscope. The percentage of cells containing autophagic vacuoles (AV) vs. the total cell number was evaluated. A minimum of 20 cells/patient were observed. Cell counting was performed by 3 independent researchers; data are presented as mean ± SD.

### 2.8. Statistical Analysis

Cell counting was conducted by 3 independent researchers, in a blinded manner. Data are presented as mean ± S.D.

To determine statistical significance, the Student’s *t*-test and Pearson’s correlation coefficient were used. Statistical significance was set at *p* < 0.05.

## 3. Results

### 3.1. Analysis of TRIM5α Levels in PBMC from LTNP vs. NP

We analyzed freshly isolated PBMC from HIV-1-infected patients and healthy donors (HD) by immunocytochemistry (Figure 1A,B). PBMC were immunolabeled, utilizing an antibody against TRIM5α, and afterwards, the TRIM5α-positive cells were quantified. A significant increase in the percentage of TRIM5α positive cells was detected in LTNP compared to NP (*p* < 0.05) (Figure 1C). Interestingly, PBMC from NP showed a significantly lower percentage of positivity also compared to HD (*p* < 0.05). In the HIV-1 controllers, more than 80% of PBMC express TRIM5α, while in the NP, fewer than 50% of cells were positive. We also analyzed the relationship existing between TRIM5α-positive cells and the viremia of patients, showing that there was no linear correlation between these parameters (Appendix A).

To confirm the above reported data, we analyzed the expression of TRIM5α by WB in PBMC from all the analyzed categories of HIV-1-infected patients and HD. The densitometric analysis showed a significantly enhanced level of TRIM5α protein in LTNP with respect to NP (*p* < 0.05). PBMC from NP also showed a significantly lower expression of TRIM5α compared to HD (*p* < 0.01) (Figure 1D).

We carried out the qPCR analysis in the analyzed categories of patients. As shown in the graph (Figure 1E), *TRIM5* mRNA showed a similar trend of expression when compared to both the immunocytochemistry and WB results, although changes were not statistically significant.

### 3.2. Quantitative Analysis of TRIM5α Expression in the PBMC Subsets

In order to identify the cell types among PBMC that expressed TRIM5α, we performed FACS analysis on PBMC from four NP, four LTNP and seven HD. TRIM5α expression on PBMC cell subsets showed a decrease on CD4 and CD8 T cells in NP and in LTNP with respect to HD (Figure 2). In contrast, in CD19 B cells and in CD14 monocytes, LTNP showed a TRIM5α expression similar to HD, while a strong decrease was observed in NP (Figure 2). These data indicate that, differently from NP, LTNP were able to maintain TRIM5α expression to healthy levels in B cells and monocytes, although the difference did not reach statistical significance in the latter cell type.

### 3.3. Colocalization of TRIM5α the HIV-1 Protein gp120 and Autophagic Vacuoles in PBMC

We performed triple immunofluorescence on PBMC from five LTNP and five NP, utilizing antibodies against TRIM5α the HIV-1 protein gp120 and LC3B (Figure 3A). We utilized the viral marker gp120 because HIV-1 assembles both at the plasma membrane of infected cells and in intracellular virus-containing compartments (VCCs), as we previously showed [23].

Interestingly, autophagic vacuoles in LTNP preferentially colocalized with the HIV-1 marker gp120. The analysis of TRIM5α intracellular distribution revealed that the protein was detected in specific areas that colocalized with both LC3B and gp120, even though it was mostly diffused in the cytosol. Quantitative analysis reported in Figure 3B showed a significantly higher percentage of colocalizations between TRIM5α, gp120, and LC3-positive dots in LTNP compared to NP (*p* < 0.0001).

Triple immunofluorescence assays have also been performed on PBMC from LTNP and NP, utilizing antibodies against the HIV-1 protein Nef together with TRIM5α and LC3B (Appendix A). In agreement with the results shown in Figure 3, autophagic vacuoles colocalized with the HIV-1 marker Nef in PBMC from LTNP.

### 3.4. Immuno-Electron Microscopy of TRIM5α and the HIV-1 Protein Nef

To obtain further information on the role of autophagy in HIV-1-infected patients, we analyzed the subcellular localization of TRIM5α and HIV-1 Nef, a protein associated with the viral core [26] by utilizing the immuno-electron microscopy on PBMC from three LTNP and three NP. Cells from NP showed HIV-1 Nef immunolabeling in vacuolar structures present in the cytoplasm, while TRIM5α labeling was mainly diffused in the cell cytosol (Figure 4A,D). The ultrastructural morphology of lymphocytes (Figure 4E) and macrophages (Figure 4F) from NP is shown.

Interestingly, the analysis of PBMC from LTNP revealed a colocalization of the analyzed proteins in vacuoles identifiable as autolysosomes (ALs). In addition, numerous gold particles were present inside ALs containing undigested material (Figure 5A–C). The ultrastructural morphology of PBMC from HIV-1 controllers, clearly showing autophagic vacuoles (AVs), were identified as a double-membrane structure containing undigested cytoplasmatic material (Figure 5E,F).

The quantitative analysis, aimed at evaluating possible differences between the HIV-1-infected patient categories, revealed that the PBMC from LTNP displayed a significantly higher percentage of cells containing AVs with respect to NP (*p* < 0.05) (Figure 5D). Moreover, the specific cells containing AVs were identified through morphological analysis, referring to the Ultrastructure Atlas of Human Tissues [27].

NP patients showed the presence of AVs in about 30% of total cells, mostly in granular leukocytes and macrophages (±75% of the total PBMC containing AVs); instead, in LTNP, AVs that were present in about 60% of total cells were identified among all the different white blood cell types recognizable as: granular leukocytes (neutrophils, eosinophils and basophils), nongranular leukocytes (lymphocytes and monocytes) and macrophages.

## 4. Discussion

In the last years, increasing evidence has shown that autophagy plays a key role in both HIV-1 replication and disease progression [6,8]. In a previous study, we demonstrated, for the first time, that in LTNP the resistance to HIV-1-induced pathogenesis is accompanied by a significant increase in the autophagic activity in PBMC [23]. Recently, the restriction factor TRIM5α has emerged as a player in the autophagy mechanism. Numerous studies have documented the association of TRIM5α with some autophagy-related proteins [13,14,15,16]; nevertheless, its effective role in viral restriction through autophagy is still debated [10,28].

Here, we investigated the hypothetical involvement of TRIM5α in controlling HIV-1 infection through autophagy, by analyzing and comparing its expression among PBMC from LTNP, NP and HD. We observed that B cells from LTNP show a significantly higher level of TRIM5α expression with respect to NP, and a positive trend is revealed in monocytes. Otherwise, CD4+ T and CD8+ T cells displayed similar TRIM5α levels in NP and in LTNP, suggesting that higher expression of TRIM5α in total PBMC from LTNP, with respect to NP, is independent from different composition of cell types typically observed in patients with different HIV disease courses. It is interesting to note that B cells and monocyte/macrophages are two subtypes notoriously not depleted during HIV-1 infection, both in NP and LTNP. In monocyte/macrophages, known as important reservoirs for the life cycle of the virus [29], the expression of TRIM5α is coupled to the expression of other autophagy-related proteins, such as BECN1, LC3s/GABARAPs, SQSTM1, HDAC6, and IRGM [8], thus emphasizing the strategic role of autophagy in the survival of these cells and viral replication during HIV-1-infection [30]. On the other hand, even B cells show higher levels of TRIM5α in LTNP, although not infected by HIV-1. It is well known that B cells show abnormalities in HIV-1-induced pathogenesis, and infected individuals show a depletion of resting memory B cells in most stages of infection, whereas nonconventional memory B cell populations are expanded, especially in HIV-viremic individuals [31]. Recently published findings showed a role for B cells in chronic constitutive inflammation and in humoral dysfunction in HIV-infected patients [32]. Although the B cell subsets were not analyzed in this study, we hypothesized that the persistence of autophagic activity in B cells of LTNP, likewise in HD, but differently from NP, seems to give more weight to the hypothesis of a ‘natural functional cure’ in this particular group of patients. The maintenance of the natural activity of B cells, including the autophagy, contributes to explaining the better immune responses against HIV-1 of LTNP. Confirming this, recent findings highlighted a role for autophagy in B cells in the regulation of humoral immune responses [33]. Overall, despite monocyte/macrophage and B cells representing a restricted percentage of total PMBC, maintenance of high expression of TRIM5α in these particular subsets of cells could be one relevant mechanism for the control of HIV infection observed in LTNP.

Interestingly, the *TRIM5α* mRNA expression level did not change significantly between the analyzed groups of HIV-1-infected patients and HD. This is in agreement with previously published data reporting that *TRIM5α* mRNA did not change significantly when comparing HIV-1-uninfected individuals, elite controllers, and ART-suppressed patients [34]. In NP patients, the TRIM5α protein expression is lower with respect to mRNA levels. A multitude of post-transcriptional, translational, and post-translational mechanisms controlling protein turnover could be involved in the stabilization of TRIM5α protein in LTNP, leading to its reduced protein degradation by the proteasomal machinery [35,36]. Further studies are required to clarify this point.

We previously showed that a number of autophagy factors, namely, BECN1, AMBRA1 and ATG5, are involved in the containment of HIV-1 in nonprogressor-infected patients [23], and here we added another autophagic pathway member probably involved in HIV-1 restriction. Moreover, previously published data showed TRIM5α in complexes with ATG14 and AMBRA1, components of the BECN1 complex engaged in autophagic initiation [12].

The colocalization of TRIM5α with HIV-1 proteins in autophagic vacuoles of LTNP suggested a role of TRIM5α in the autophagic restriction of HIV-1 particles. This evidence is supported by the notion that HIV-1 can be degraded into autophagosomes in LTNP [23]. Moreover, TRIM5α has been previously shown to act as a selective autophagy receptor, for the recognition of viral capsid protein and initiation of autophagy [12]. In its role as an autophagic cargo receptor, TRIM5α directly recognizes viral capsid sequences via its SPRY domain [18,19]. The localization of TRIM5α and HIV-1 proteins inside autophagic vacuoles in LTNP is in agreement with very recently published findings, showing that TRIM5α can form hexagonal nets inside cells, as well as different structures whose features match those expected for different stages of macroautophagy [17].

HIV-1 restriction by TRIM5α through autophagy has also been demonstrated in human dendritic cell subsets, where it mediates the assembly of an autophagy-activating scaffold which targets HIV-1 for autophagic degradation [14]. Recently, it has also been shown that autophagy machinery provides TRIM5α with a platform upon which to activate anti-retroviral responses [22].

In conclusion, the results reported in this paper suggest the participation of TRIM5α in the autophagic containment of HIV-1 in LTNP. In our opinion, modulation of the autophagy mechanism can be considered an attractive therapeutic target in the fight against HIV-1 infection. Indeed, understanding the cellular and molecular mechanisms at the basis of the successful control of the chronic viral infection in vivo, in HIV-1-controllers, it represents a very promising avenue for the development of novel therapeutic strategies to treat HIV-1 infection. Moreover, many autophagy inducers are currently under clinical investigation for the treatment of several diseases [37,38,39,40], and they might be employed in combination with antiretroviral drugs administered to HIV-1-infected normal progressor patients in order to improve the treatment of HIV-1 infection.

## Figures and Tables

**Figure 1 cells-10-01207-f001:**
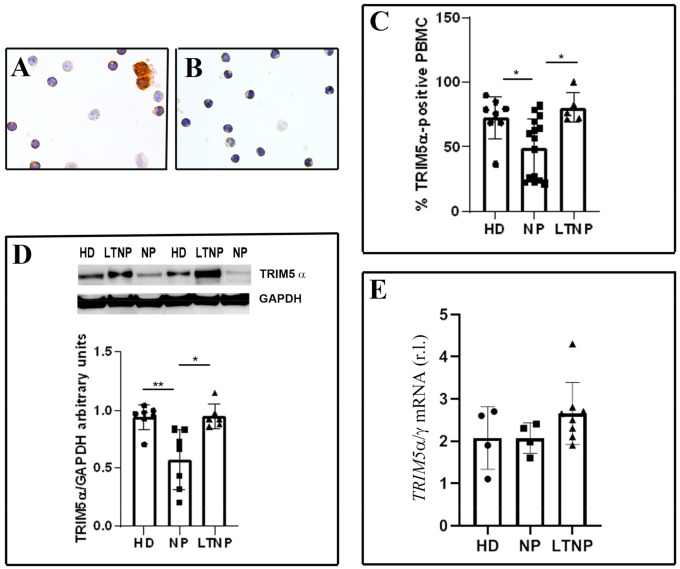
Enhanced expression of TRIM5α in PBMC from LTNP compared to NP. (**A**) Immunocytochemical localization of TRIM5α on PBMC from LTNP (Patient No. 3). (**B**) Negative control. (**C**) The percentage of cells expressing TRIM5α was quantified on PBMC from 15 NP (All enrolled patients), 5 LTNP (Patients No. 1, 2, 4, 6, and 9) and 8 HD, and a minimum of 500 PBMC/patient were analyzed. (**D**) WB analysis of TRIM5α levels on PBMC from 7 HD, 7 NP (Patients No. 4, 6, 8, 9, 13, 14, and 15) and 6 LTNP (Patients No. 1, 2, 3, 4, 7, and 10). The mean values of 3 different experiments are reported in the graph. (**E**) qPCR of *TRIM5* mRNA relative level on PBMC from 4 NP (Patients No. 3, 5, 12, and 14), 8 LTNP (Patients No. 1–5, 7, 9, and 10) and 4 HD. The mean values of 4 different experiments are reported in the graph. Data are reported as scatter plots with means values ± s.d. * *p* < 0.05, ** *p* < 0.01. HD, healthy donors; NP, HIV-1-infected normal progressor patients; LTNP, long-term nonprogressors. Original magnification: A, B, 40×.

**Figure 2 cells-10-01207-f002:**
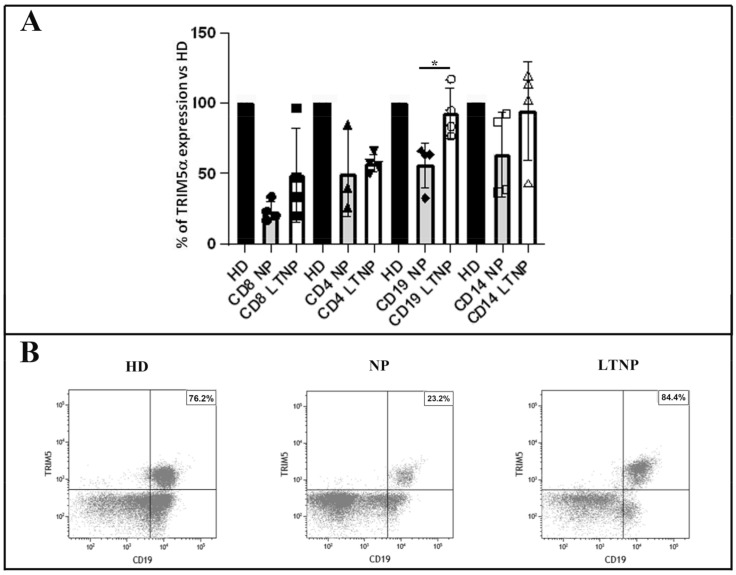
Expression of TRIM5α in PBMC cell populations. (**A**) The graphs show TRIM5α expression in specific PBMC cell subpopulations, namely, T CD4, T CD8, monocytes CD14 and B CD19 cells. Data have been obtained from 7 HD, 4 NP (Patients No. 2, 5, 11, and 14) and 4 LTNP (Patients No. 1, 5, 8, and 9) normalized to HD. In the graph, data are reported as scatter plots with means values ± s.d. * *p* < 0.05. (**B**) Representative flow cytometric panels showing TRIM5α+ expressing B cells (CD19+) in HD, LNTP and NP. Percentage of TRIM5α+ expressing CD19+ cells among total B cells are depicted in each panel (upper-right quadrant). PBMC were first gated including only singlet cells, then as live cells (7AAD-), and finally, among CD3-negative cells, CD19+ TRIM5α+ cells were selected. HD, healthy donors; NP, HIV-1-infected normal progressor patients; LTNP, long-term nonprogressors.

**Figure 3 cells-10-01207-f003:**
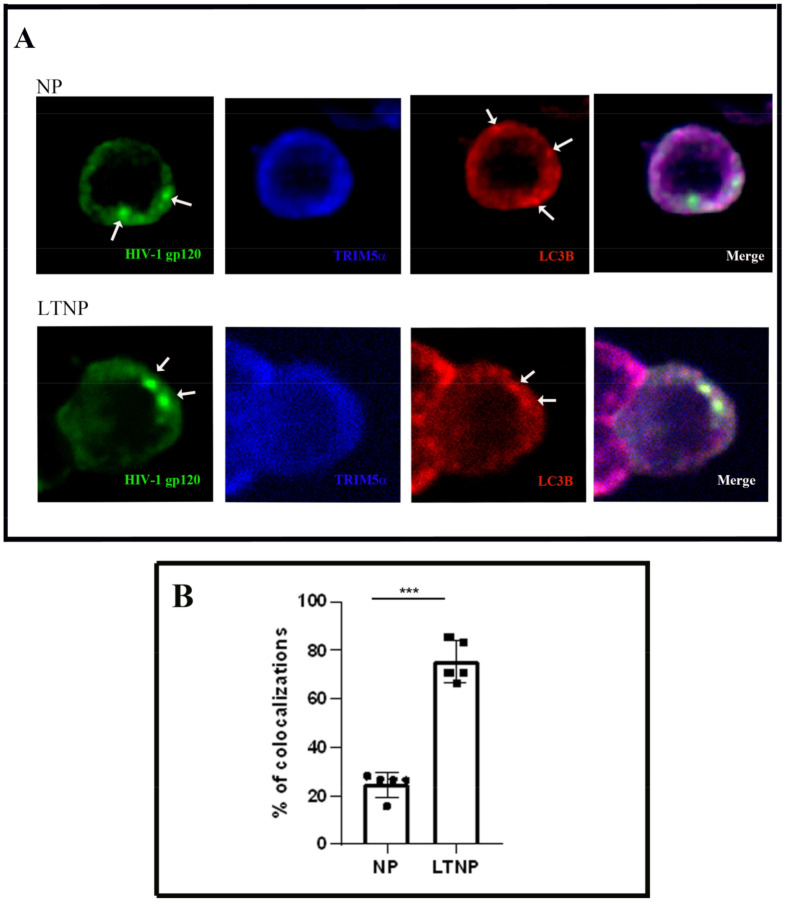
Colocalization of HIV-1 gp120 protein and TRIM5α in autophagic vacuoles in PBMC from LTNP. (**A**) Confocal microscopy immunolocalization of the HIV-1 protein gp120 (green), TRIM5α (blue), and the autophagosome marker LC3B (red) on PBMC from 5 LTNP (Patients No. 1, 2, 4, 5, and 10) and 5 NP (Patients No. 3, 5, 6, 7, 8). The white arrows indicate the proteins dots. (**B**) Quantitative analysis of the percentage of colocalization between HIV-1 protein gp120, LC3B and TRIM5α with respect to the HIV-1 gp120 dots. In the graph are the mean values of immunolocalizations performed on PBMC from NP and LTNP. Data are reported as scatter plots with mean values ± s.d. *** *p* < 0.001. NP, HIV-1-infected normal progressor patients; LTNP, long-term nonprogressors. Original magnification: A, 63×.

**Figure 4 cells-10-01207-f004:**
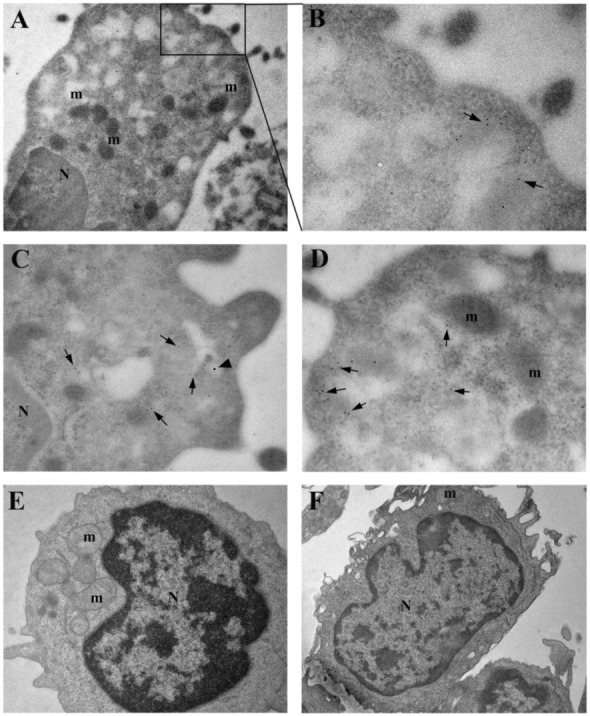
Immuno-electron microscopy and ultrastructural analysis of PBMC from HIV-1-infected NP patients. (**A**,**D**) PBMC from 5 NP (Patients No. 2, 9, 10, 12, and 15) were subjected to immunogold analysis. A double immunolabeling was performed, utilizing anti-TRIM5α (15 nm gold particles, arrowheads) and anti-HIV-1 Nef (5 nm gold particles, arrows) antibodies. (**A**) Numerous 5 nm gold particles (arrows) labeling HIV-1 Nef are present in vacuole-enriched areas of the cytoplasm of a macrophage. (**B**) Magnification of the boxed area in A. (**C**,**D**) Arrowheads indicate that TRIM5α immunolabelling is present in the cytoplasm. Ultrastructural morphology of a lymphocyte (**E**) and of a macrophage (**F**) from HIV-1-infected NP patients. N, nucleus; m, mitochondria. Original magnifications: A, 15,000×; B–D, 50,000×; E, 20,000×; F, 12,000×.

**Figure 5 cells-10-01207-f005:**
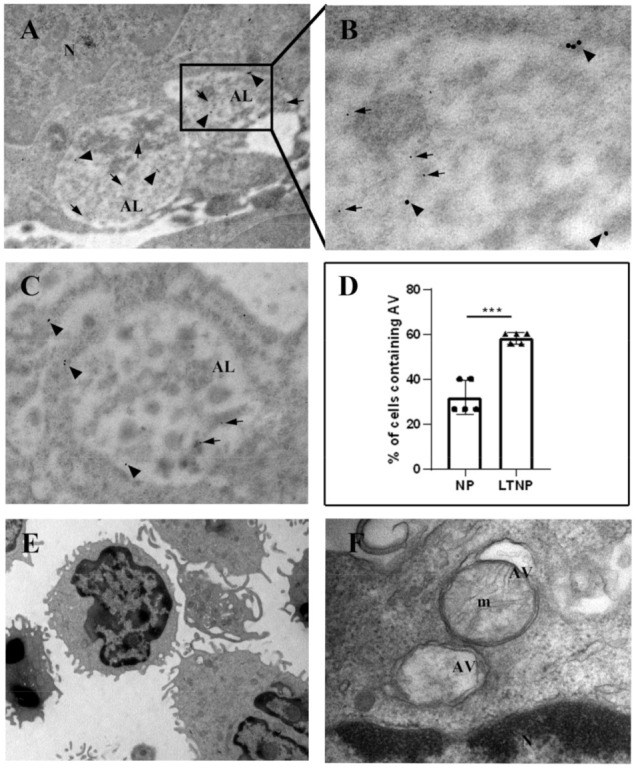
Immuno-electron microscopy and ultrastructural analysis of PBMC from HIV-1-infected LTNP patients. (**A**–**D**) PBMC from 5 LTNP (Patients No. 1, 3, 6, 8, and 10) were subjected to immunogold analysis. HIV-1 Nef (arrows) and TRIM5α (arrowheads) immunolabelling colocalized in ALs. The images show autolysosomes (ALs) containing undigested material, in which gold particles labelling HIV-1 Nef (arrows) and TRIM5α (arrowheads) are present. (**B**) Higher magnification of the boxed area presented in Figure 4A. (**C**) AL is visible, containing gold particles of 5 and 15 nm. (**D**) Quantitative analysis of the percentage of cells containing AV in 5 NP and 5 LTNP. In the graph, data are reported as scatter plots with mean values ± s.d. *** *p* < 0.001. (**E**) Ultrastructural analysis of PBMC from HIV-1 controllers. (**F**) AVs surrounded by double membranes. AL, autolysosome; AV, autophagic vacuole; N, nucleus; m, mitochondria. NP, HIV-1-infected normal progressor patients; LTNP, long-term nonprogressors. Original magnifications: A, 20,000×, B, 85,000×; C, 50,000×; E, 7000×; F, 50,000×.

## Data Availability

The data presented in this study are available on request from the corresponding author.

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
