# Peer review of "High Levels of TRIM5α Are Associated with Xenophagy in HIV-1-Infected Long-Term Nonprogressors"

_cells, 2021, doi:10.3390/cells10051207_

Round 1

Reviewer 1 Report

In their revision of manuscript cells-1189807, the authors have answered all my earlier remarks and have adapted the text accordingly. Before publication, I would advise adding two minor explanations to the Materials & Methods section on RT-PCR & qPCR, namely that L34 is a ribosomal protein, and that the TRIM5alpha and gamma isoforms cannot be distinguished with the PCR used

Author Response

REVIEWER 1

Open Review

English language and style

( ) Extensive editing of English language and style required
(x) Moderate English changes required
( ) English language and style are fine/minor spell check required
( ) I don't feel qualified to judge about the English language and style

Yes

Can be improved

Must be improved

Not applicable

Does the introduction provide sufficient background and include all relevant references?

(x)

( )

( )

( )

Is the research design appropriate?

(x)

( )

( )

( )

Are the methods adequately described?

(x)

( )

( )

( )

Are the results clearly presented?

(x)

( )

( )

( )

Are the conclusions supported by the results?

(x)

( )

( )

( )

Comments and Suggestions for Authors

In their revision of manuscript cells-1189807, the authors have answered all my earlier remarks and have adapted the text accordingly.

Before publication, I would advise adding two minor explanations to the Materials & Methods section on RT-PCR & qPCR, namely that L34 is a ribosomal protein, and that the TRIM5alpha and gamma isoforms cannot be distinguished with the PCR used

We thank the Reviewer for his/her comments, we added the requested explanations to the Materials & Methods section, on page 4, highlighted by "Track Changes”.

Reviewer 2 Report

The paper

TRIM5α participates in the autophagy-dependent control of HIV-1 in long-term non progressor patients

shows great improvement compared to the previous version.

Despite this, still many concerns are present that can impact the study publication.

The title of the study clearly states that TRIM5a participate in the control of HIV-1 in LNTP but this is not supported from the actual data.

It is well known that hTRIM5 can bind HIV but it fails in the degradation process; at the contrary rhTRIM5 can perform this reaction. Actually a single amino-acid change (P332R) is sufficient to lead to HIV-1 restriction by human TRIM5α.

The association of hTRIM5 and HIV in the cytoplasmic bodies (or ALs) doesn’t  prove HIV restriction (microscopy data) in any way. (Functional experiments, iRNA for example, would be more appropriated).

Peculiar are the data in fig 2. The means (bars) present in the original submission are different of the ones present in the last submission (CD4 and CD8 in particular) , despite it is not stated anywhere any difference in donor’s number between the two versions.

In fig 1C the differences between the different groups look more driven by a different component of NP (low values), without this the difference would be not significative. But it is difficult to compare the whole fig1 :

Fig 1C LNTP  5  NP  15   HD  8

Fig 1D LNTP  6  NP 7     HD 7

Fig 1E LNTP  8  NP  4     HD  4

Do the LNTP used in the 3 analysis are the same? Do the NP in D and E are a subset of C or different ones???

In any case what are the conclusions for this experiment? It looks like TRIM5 protein in NP has a faster turn over compared to LNTP and HD (same RNA amount), but this does not prove restriction activity in LNTP.

Comparing the TRIM5 expression between fig1 and fig2 again :

Fig 1C LNTP  5  NP  15   HD  8

Fig 2A LNTP  4  NP 4   HD  7

Are this the same samples? Or different ones? How the subset was decided?

In fig1C NP-PBMNC have a lower content of Trim5 but in fig2 this is true just for CD19, that are not HIV targets!

In brief, the data don t support the claim that in LNTP TRIM5 is responsible (or participate) in HIV control.

The experiment design must be improved.

Author Response

REVIEWER  2

Open Review

English language and style

( ) Extensive editing of English language and style required
( ) Moderate English changes required
( ) English language and style are fine/minor spell check required
(x) I don't feel qualified to judge about the English language and style

Yes

Can be improved

Must be improved

Not applicable

Does the introduction provide sufficient background and include all relevant references?

(x)

( )

( )

( )

Is the research design appropriate?

( )

(x)

( )

( )

Are the methods adequately described?

( )

(x)

( )

( )

Are the results clearly presented?

( )

(x)

( )

( )

The paper

TRIM5α participates in the autophagy-dependent control of HIV-1 in long-term non progressor patients

shows great improvement compared to the previous version.

Despite this, still many concerns are present that can impact the study publication.

The title of the study clearly states that TRIM5a participate in the control of HIV-1 in LNTP but this is not supported from the actual data.

Taking into account the Reviewer’s criticisms and in agreement with the Referee’s suggestion, we modified the manuscript title as follows:

“High levels of TRIM5a are associated with xenophagy in HIV-1- infected long-term nonprogressors”

It is well known that hTRIM5 can bind HIV but it fails in the degradation process; at the contrary rhTRIM5 can perform this reaction. Actually a single amino-acid change (P332R) is sufficient to lead to HIV-1 restriction by human TRIM5α.

 It is known that while rhesus TRIM5α potently restricts HIV-1 infection, human TRIM5α is only weakly active against HIV-1.

However, human TRIM5α has been described to block HIV-1 infection in Langerhans cells (C. M. Ribeiro et al., Receptor usage dictates HIV-1 restriction by human TRIM5α in dendritic cell subsets. Nature 540, 448–452 (2016) mediating the assembly of an autophagy-activating scaffold to Langerin, which targets HIV-1 for autophagic degradation and prevents infection of  Langerhans cells. Data from this study suggest that autophagy induction alone does not mediate human TRIM5α-dependent restriction of HIV-1, but that it requires the formation of a complex between HIV-1, human TRIM5α and autophagy molecules for restriction. Mandell MA and colleagues described TRIM5α as an autophagic receptor governing selective autophagy of HIV-1 capsid (Mandell, M.A.; et al. TRIM proteins regulate autophagy and can target autophagic substrates by direct recognition. Dev Cell 2014, 30, 394–409).

Of course, the full characterization of the mechanism/s through which TRIM5α restricts HIV-1 is still open and needs to be further investigated.

The association of hTRIM5 and HIV in the cytoplasmic bodies (or ALs) doesn’t  prove HIV restriction (microscopy data) in any way. (Functional experiments, iRNA for example, would be more appropriated).

 We agree with the Referee comment that only functional experiments can prove the effective HIV-1 restriction by TRIM5α, but we based our conclusions on previously published functional data, obtained by in vitro experiments, demonstrating that TRIM5 directly recognizes its cognate retrovirus capsid and orchestrates its autophagic degradation, thus protecting cells against HIV-1 infection (Mandell MA et al. Dev Cell. 2014 Aug 25;30(4):394-409; Mandell MA et al. Autophagy. 2014 Dec; 10(12): 2387–2388). 

We are aware that data shown in this study are descriptive, but our results are obtained in vivo, by analyzing rare HIV-infected subjects such as LTNP, and in our opinion this information may be relevant for the scientific community. Taking into account the study limit and the Reviewer criticism, we revised the Discussion section and the manuscript Title, in order to focus on this aspect.  

Peculiar are the data in fig 2. The means (bars) present in the original submission are different of the ones present in the last submission (CD4 and CD8 in particular) , despite it is not stated anywhere any difference in donor’s number between the two versions.

Data showed in the revised version of the paper are different from the original submission since we performed additional experiments in order to carry out a statistical analysis, as requested by Reviewer 1.

PBMC from 6 HD, 2 NP and 3 LTNP were utilized to obtain data showed in the original submission and PBMC from 7 HD, 4 NP and 4 LTNP were utilized to obtain data showed in revised version. For this experiment, we utilized frozen PBMC from previously enrolled donors.

In fig 1C the differences between the different groups look more driven by a different component of NP (low values), without this the difference would be not significative. But it is difficult to compare the whole fig1 :

 Fig 1C LNTP  5  NP  15   HD  8

Fig 1D LNTP  6  NP 7     HD 7

Fig 1E LNTP  8  NP  4     HD  4

 Do the LNTP used in the 3 analysis are the same?

The LTNP patients enrolled for immunohistochemistry were the same utilized in the other analyses, of course  1 more patient was enrolled for western blot analysis and 3 more patients were additionally included for  RT-PCR & qPCR.

Do the NP in D and E are a subset of C or different ones???

The NP patients enrolled for western blot analysis (Figure 1D) and for RT-PCR & qPCR (Figure 1E) were a subset of patients enrolled for immunocytochemical analysis (Figure 1C).

In any case what are the conclusions for this experiment? It looks like TRIM5 protein in NP has a faster turn over compared to LNTP and HD (same RNA amount), but this does not prove restriction activity in LNTP.

As correctly noted by the Reviewer, the TRIM5α mRNA expression level did not change significantly between the analyzed groups, accordingly as reported by other Authors (Abdel-Mohsen M, et al. (2013) Expression profile of host restriction factors in HIV-1 elite controllers. Retrovirology 10: 106). A multitude of post-transcriptional and posttranslational mechanisms controlling protein turnover could be involved in the stabilization of the protein in LTNP, further studies are required to clarify this point.

Of course this experiment does not prove restriction activity in LNTP but it proves that TRIM5α protein levels are higher in LTNP.  

It is interesting to note that the higher expression of TRIM5α in LTNP respect to NP is comparable to the expression of some autophagy factors, namely BECN1, AMBRA1 and ATG5, that are involved in the containment of HIV-1 in nonprogressor-infected patients [Nardacci, R.; et al. Autophagy plays an important role in the containment of HIV-1 in nonprogressor-infected patients. Autophagy 2014, 10, 1167-78].

Cells from LTNP show high levels of TRIM5α expression respect to NP, particularly in monocytes and B cells that represent a restricted percentage of total PMBC, but the maintenance of high expression of TRIM5α in these particular subsets of cells could be a relevant mechanism for the control of HIV infection observed in LTNP. Moreover, the maintenance of high expression of TRIM5α in these cell subsets, particularly in monocyte/macrophages, is coupled to the expression of other autophagy-related proteins, such as BECN1, LC3s/GABARAPs, SQSTM1, HDAC6, and IRGM (Nardacci, R. et al.. Role of autophagy in HIV infection and pathogenesis. J Intern Med 2017; 281, 422-432), thus emphasizing the strategic role of autophagy in the survival of these cells and viral replication during HIV-1-infection (Espert, Let al. Differential role of autophagy in CD4 T cells and macrophages during X4 and R5 HIV-1 infection. PLoS One. 2009, 4, e5787).

We understand the Reviewer’s criticism; in agreement we revised the Discussion section and the manuscript Title.

Comparing the TRIM5 expression between fig1 and fig2 again :

 Fig 1C LNTP  5  NP  15   HD  8

Fig 2A LNTP  4  NP 4   HD  7

Are this the same samples? Or different ones? How the subset was decided?

The patients enrolled for FACS analysis (Figure 2) were a subset of patients enrolled for immunocytochemical analysis (Figure 1C) .

We comprehend the reviewer's criticism of the low number of patients enrolled, but LTNP are a small proportion of infected individuals (5% to 15%). We could only recruit 10 LTNP patient, accordingly we enrolled a comparable number of patients belonging to the other categories.

So far, a limited number of studies have been carried out on LTNP patients, due to the relative low number of subjects. Thus, despite the debatable low number of patients enrolled, we think that data shown in this paper may be of interest to the scientific community.

In fig1C NP-PBMNC have a lower content of Trim5 but in fig2 this is true just for CD19, that are not HIV targets!

 We answer to a similar question in the first revision of our manuscript. Since the Editorial Office may have send the revised paper to a new Reviewer, we attach the previous question and the relative answer.

Furthermore, from Fig 2 it is also apparent that TRIM5α levels in CD4 and CD8 T cells, the most relevant cells for HIV-1 infection, are significant lower in all HIV-1 infected individuals compared with HD. How biologically significant is it for disease progression to have comparable levels of TRIM5α in B cells, when B cells are not infected by HIV-1? Please explain

We thank the Referee's comment which allowed us to clarify this point in the Discussion section, as follows:

 On the other hand, even B cells show higher levels of TRIM5α in LTNP, although not infected by HIV-1. It has been demonstrated that people with HIV-1 possess an unusual subset of exhausted, non-responsive cells called tissue-like B-memory cells, which seem to contribute to the inefficiency of HIV-1-specific antibody responses [Kardava L et al. Attenuation of HIV-associated human B cell exhaustion by siRNA downregulation of inhibitory receptors. Journal of Clinical Investigation, early online edition doi: 10.1172/JCI45685. 2011]. The persistence of autophagic activity in B cells of LTNP, likewise in HD, but differently from NP, seems to give more weight to the hypothesis of a ‘natural functional cure’ in these particular group of patients, occurring through the maintenance of the natural activity of B cells, including the autophagy, thus contributing to explain the better immune responses against HIV-1. Confirming this, recent findings highlighted a role for autophagy in B cells in the regulation of humoral immune response [He C, et al. CD36 and LC3B initiated autophagy in B cells regulates the humoral immune response. Autophagy. 2021, Feb 3].

In brief, the data don t support the claim that in LNTP TRIM5 is responsible (or participate) in HIV control.

The experiment design must be improved.

In consideration of the Reviewer’s criticisms, and taking into account that in this manuscript we did not state that a direct action of TRIM5α is the main factor involved in disease progression, we decided to change the manuscript Title and to revise the Discussion section accordingly.

Round 2

Reviewer 2 Report

I would suggest to present the figures as color/number coded-donor dependent results. Moreover a detailed table of the clinical information of the LNTP color/number coded would be very opportune. This to be able to associate the clinical information of every donor in the different figures.

About your answers:

Cells from LTNP show high levels of TRIM5α expression respect to NP, particularly in monocytes and B cells that represent ........

Moreover, the maintenance of high expression of TRIM5α in these cell subsets, particularly in monocyte/macrophages, is coupled to the expression of other autophagy-related proteins, such as BECN1, LC3s/GABARAPs,.....

I would like to point it out that you don t have a statistical significant difference in TRIM expression in the CD14 population. At best it is a positive trend.

About your answers:

It has been demonstrated that people with HIV-1 possess an unusual subset of exhausted, non-responsive cells called tissue-like B-memory cells, which seem to contribute.......

Confirming this, recent findings highlighted a role for autophagy in B cells in the regulation of humoral immune ......

it is well known that there is a expansion of tissue-like B-memory cells in HIV for example

https://doi.org/10.1016/j.ijid.2016.02.108

DOI: 10.1128/JVI.00298-12

https://doi.org/10.1038/s41590-018-0180-5   and the paper CD36-autophagy (in mice) is for sure very intriguing, but you did not subset your B cells population so it is difficult to find a strong link with your study.   I think the change of the title was very opportune.    

Author Response

Point-by-point response

 REVIEWER 2

I would suggest to present the figures as color/number coded-donor dependent results. Moreover a detailed table of the clinical information of the LNTP color/number coded would be very opportune. This to be able to associate the clinical information of every donor in the different figures.

Accordingly with the Reviewer’s suggestion we added Supplementary Table 1 (LTNP patients) and Supplementary Table 2 (NP patients) summarizing the demographic and laboratory findings of our patient cohort. Numbers, coding the individual patients, have been indicated in each Figure legend, in order to associate patient’s clinical information to the showed results.

In the Materials and Methods section, the median values of plasma HIV-1 RNA (viral load) have been expressed homogeneously and updated, considering the patients enrolled for the experiments carried out during the review process.

About your answers:

Cells from LTNP show high levels of TRIM5α expression respect to NP, particularly in monocytes and B cells that represent ........

Moreover, the maintenance of high expression of TRIM5α in these cell subsets, particularly in monocyte/macrophages, is coupled to the expression of other autophagy-related proteins, such as BECN1, LC3s/GABARAPs,.....

I would like to point it out that you don t have a statistical significant difference in TRIM expression in the CD14 population. At best it is a positive trend.

 In agreement with the Reviewer’s criticisms, we revised the text in the Discussion sections as follows:

“We observed that B cells from LTNP show a significantly higher level of TRIM5α expression respect to NP and a positive trend is revealed in monocytes”

“In monocyte/macrophages, known as important reservoir for the life cycle of the virus [29], the expression of TRIM5α is coupled to the expression of ….”

Modifications have been highlighted by "Track Changes”.

About your answers:

It has been demonstrated that people with HIV-1 possess an unusual subset of exhausted, non-responsive cells called tissue-like B-memory cells, which seem to contribute.......

Confirming this, recent findings highlighted a role for autophagy in B cells in the regulation of humoral immune ......

it is well known that there is a expansion of tissue-like B-memory cells in HIV for example

https://doi.org/10.1016/j.ijid.2016.02.108

DOI: 10.1128/JVI.00298-12

https://doi.org/10.1038/s41590-018-0180-5   and the paper CD36-autophagy (in mice) is for sure very intriguing, but you did not subset your B cells population so it is difficult to find a strong link with your study.   I think the change of the title was very opportune.    

We understand the Reviewer’s criticism about B cells. We observed that CD19+ cells expressed the same levels of TRIM5alpha as those in HD, thus suggesting a possible connection between this evidence and the health of LTNP.

In this study naïve and resting, activated, and tissue-like memory subpopulations of B cells were not evaluated individually for TRIM5alpha expression. Of course, further studies are needed in order to explore TRIM5alpha expression in the various subpopulations of CD19+ cells.

In agreement with the Reviewer’s comment, we revised the Discussion section as follows:

“It is well known that B cells showed abnormalities in HIV-1-induced pathogenesis, infected individuals show depletion of resting memory B cells in most stages of infection, whereas nonconventional memory B cell populations are expanded, especially in HIV-viremic individuals (Moir, S. et al. B cells in early and chronic HIV infection: evidence for preservation of immune function associated with early initiation of antiretroviral therapy. Blood 116, 5571–5579.2010).  Recently published findings showed a role for B cells in chronic constitutive inflammation and in humoral dysfunction in HIV-infected patients (Leal, V.N.C.; Reis, E.C., Fernandes, F.P.; Soares, J.L.D.S.; Oliveira, I.G.C.; Souza de Lima, D.; Lara, A.N.; Lopes, M.H.; Pontillo, A. Common pathogen-associated molecular patterns induce the hyper-activation of NLRP3 inflammasome in circulating B lymphocytes of HIV-infected individuals.AIDS. 2021,35, 899-910). Although the B cell subsets were not analyzed in this study, we hypothesized that the persistence of autophagic activity in B cells of LTNP, likewise in HD, but differently from NP, supports the hypothesis of a ‘natural functional cure’ in these particular group of patients. The maintenance of the natural activity of B cells, including autophagy, contributes to explain the better immune responses against HIV-1 of LTNP”.

This manuscript is a resubmission of an earlier submission. The following is a list of the peer review reports and author responses from that submission.

Round 1

Reviewer 1 Report

LNTP and EC are interesting subjects because they are able to naturally control HIV infection.

There has been a plethora of studies trying to define the determinants for such protection with sparse results. Indeed, LNTP and EC are not homogenous groups and the resilience to HIV infection has been attributed to multiple and different factors that are donor or virus dependent.

In this study, Dr Ciccosanti & co. is suggesting a role of TRIM5alpha in controlling the viral infection in LNTP group compared to the NP group.

This study shows that PBMC from NP have less TRIM5alpha compared to HD and LNTP.

CD4 and CD8 both have lower level of TRIM5alpha in NP and LNTP compared to HD

CD19 and CD14 both have less TRIM5alpha in NP compared to LNTP and HD

Immunofluorescence and immuno-electron microscopy both show a positive action of TRIM5alpha in binding and degrading HIV proteins.

The study is interesting, but it must further improved.

Fig 1 shows that NP have less TRIM5alpha despite the RNA levels for TRIM5alpha in HD, NP, and LNTP are identical. It would be important to understand the reason of this pattern.

In Fig 2, the level of TRIM5alpha are quantified by FACS in the subpopulation CD4, CD8, CD14, CD19. No standard deviations are present, suggesting that this experiment was performed just once. This is in general not a good approach. In particular, the difference in TRIM5alpha expression in NP and LNTP in CD14 is quite similar and not informative without SD/p-value.

Immunofluorescence and Immuno-electron microscopy should be performed on specific immune cell populations (CD4, CD14 for example)

The authors in the discussion point out the role of CD14 as reservoir for HIV but this is still a highly debated subject with no definitive consensus.

LNTP are not a homogenous group and it would be important to try to correlate clinical determinates (example: residual viral load, size of reservoir) with the TRIM5alpha level.

In brief, despite the study offers some interesting information, the experimental part should be improved and expanded and the discussion is not fully supported by the data.

A general rethinking of the study is opportune.

Reviewer 2 Report

The manuscript by Ciccosanti et al concerns the analysis of TRIM5α expression in HIV-1 infected normal progressors (NP), long-term nonprogressors (LTNP) and healthy controls (HD).

The authors show that TRIM5α levels are normal in LTNP, but not in NP, and that TRIM5α is associated with autophagy in HIV-1 infection, although they state that levels in NP are lower than in LTNP, from Figs. 1 and 2 it appears that TRIM5α levels in LTNP are similar to HD, while those in NP are lower. So, I would rather put it that way. Furthermore, from Fig 2 it is also apparent that TRIM5α levels in CD4 and CD8 T cells, the most relevant cells for HIV-1 infection, are significant lower in all HIV-1 infected individuals compared with HD. How biologically significant is it for disease progression to have comparable levels of TRIM5α in B cells, when B cells are not infected by HIV-1? Please explain.

Fig. 1, panels D and E: How were TRIM5α levels determined? What are ‘arbitrary units’ of GAPDH? And what is gamma mRNA? In the latter comparison, there is no difference between NP and LTNP in TRIM5α expression, refuting your hypothesis. Please explain.

And why have NP and LTNP, with a different disease course, similar low levels of TRIM5α in T cells, which would suggest that a direct action of TRIM5α is not the main factor involved in disease progression? Please explain.

Next, the choice of HIV-1 protein in the experiments shown in Fig. 3 is puzzling. TRIM proteins have been shown to exert their antiviral activity through retroviral capsid binding. But here, evidence for interaction with HIV-1, resulting in co-localisation in the autophagic vacuole, is given by staining the envelope protein gp120. Would that imply that TRIM5α directs newly formed viral proteins to the autophagosome, as well as interfering with uncoating? The envelope proteins are left outside the cell after membrane fusion of the virion with the host cell, while the capsid containing the viral genome enters the cytoplasm to be met by restriction factors such as TRIM proteins. Why not use for instance capsid-p24? Please explain.

Also, Figs 4 and 5 show results obtained with PBMC from NP, or LTNP, respectively. Could you be more specific as to the cell type shown here? Comparing for instance a B cell with a T cell would not make much sense, and it looks like a single cell has been used in the panels.

Furthermore, could there be sequence variation in the HIV capsid responsible for the difference between NP and LTNP? Or, were any TRIM5α polymorphisms present between the two HIV-1 infected groups? For instance G249D (see: Kömürlü et al., PLoS ONE, 2019)?

Minor comments:

Overall, conversion of the file to MDPI format apparently resulted in multiple errors, for instance the alpha is often missing in TRIM5alpha. Please check throughout. Furthermore, English editing and a careful check of captions to the axes of the figures is necessary throughout the manuscript.

Line 23: Reads awkward; ‘it has been identified’ is not correct

Lines 70-73: letter size too small

Line 78, 80: something is missing in the description of CD4+ T cell levels

Legend Fig. 2: alpha is missing from TRIM5

Legend Fig. 3: ‘blue’, not ‘blu’. Please explain why LC3B is stained here.

In figure legends: ‘are shown’, not ‘were reported’

Line 265: what does ‘that results diffuse in the cytoplasm’ mean?

Lines 258, 271: ‘shown’, not ‘showed’

Line 298: TCD4 and TCD8 cells are not commonly used abbreviations